# Detection of *Plasmodium falciparum* in Saliva and Stool Samples from Children Living in Franceville, a Highly Endemic Region of Gabon

**DOI:** 10.3390/diagnostics13203271

**Published:** 2023-10-20

**Authors:** Roméo Karl Imboumy-Limoukou, Jean-Claude Biteghe-Bi-Essone, Judicael Boris Lendongo Wombo, Sonia Etenna Lekana-Douki, Virginie Rougeron, Steede-Seinnat Ontoua, Lydie Sandrine Oyegue-Liabagui, Cherone Nancy Mbani Mpega Ntigui, Lady Charlène Kouna, Jean-Bernard Lekana-Douki

**Affiliations:** 1Unité Evolution Epidémiologie et Résistance Parasitaire (UNEEREP), Centre International de Recherches Médicales de Franceville (CIRMF), Franceville BP 769, Gabon; biteghebiteghe@gmail.com (J.-C.B.-B.-E.); borislendongo@yahoo.fr (J.B.L.W.); ontouaseinnat@gmail.com (S.-S.O.); lyds_ass@yahoo.fr (L.S.O.-L.); mpega_mb2@yahoo.fr (C.N.M.M.N.); charleneklc@gmail.com (L.C.K.); lekana_jb@yahoo.fr (J.-B.L.-D.); 2Laboratoire de Biologie Moléculaire et Cellulaire (LABMC), Université des Sciences et Techniques de Masuku, Franceville BP 943, Gabon; 3Unité des Maladies Virales Emergentes (UMVE), Centre International de Recherches Médicales de Franceville, Franceville BP 769, Gabon; s_lekana@yahoo.fr; 4MIVEGEC, IRD, CNRS, University of Montpellier, 34900 Montpellier, France; rougeron.virginie@gmail.com; 5Ecole Doctoral Régional en Infectiologie Tropical, Franceville BP 876, Gabon; 6Département de Parasitologie-Mycologie, Université des Sciences de la Santé, Libreville BP 4008, Gabon

**Keywords:** malaria diagnosis, *Plasmodium falciparum*, saliva, stools, qPCR *STEVOR*

## Abstract

Due to the difficulty of obtaining blood samples, which is the invasive method that is currently used for the detection of *Plasmodium* spp., alternative diagnostic sampling methods that are effective and non-invasive are needed, particularly for long-term studies. Saliva and stool samples from malaria-infected individuals contain trace amounts of *Plasmodium* DNA and therefore could be used as alternatives. Malaria was screened using rapid diagnosis tests and confirmed via microscopy. Nested PCR tests targeting the *Plasmodium falciparum*-specific *STEVOR* gene were performed for blood, saliva and stool samples that were positive for malaria. Three hundred sixty-seven (367) children were enrolled and eighty (22.22%) were confirmed to be positive for malaria. Matched blood, saliva and stool samples were available for 35 children. By using blood smears as the gold standard for the diagnosis of malaria, our study indicates that *Plasmodium* DNA was more detectable in blood (100%) than in saliva (22.86%) and stools (14.29%). Applying qPCR to the *STEVOR* gene to detect *Plasmodium falciparum* DNA in saliva and stool samples cannot be considered as an alternative to the current malaria detection processes using blood specimens.

## 1. Introduction

Malaria continues to pose a huge public health threat globally. In 2020, malaria accounted for 627,000 deaths in the world, and 95% of the deaths occurred in sub-Saharan Africa [1]. Malaria is the result of a *Plasmodium* infection, which, if left untreated, can progress from asymptomatic parasitaemia to uncomplicated malaria, and then to severe malaria, and finally, death. Practically in all endemic countries, a significant decline in disease rates has been demonstrated, which is probably due to the large-scale bed net program and the improvement in case management such as enhanced diagnostic tests and the implementation of highly effective anti-malarial drugs. Clinical malaria is characterised by the presence of a malaria parasite in the blood, and is associated with other symptoms such as intermittent fever, malaise, shaking, chills, arthralgia, myalgia, vomiting, etc. [2]. 

In Gabon, a Central African country, the malaria transmission rate is stable throughout the year, with a mean prevalence of 22.3% (according to the data from the National Malaria Control Programme) due to its warm and humid equatorial climate, which favours the proliferation of mosquitos [3]. In the country, three plasmodial species, *P. falciparum*, *P. malariae* and *P. ovale*, have been reported, and *P. falciparum* represents 95% of the encountered isolates [4]. In Gabon, the control of malaria remains a major public health challenge, especially in both pregnant women and children. Several studies have been carried out to improve the understanding of the characteristics of infection in this vulnerable population [5,6]. In Franceville (in southeast Gabon), malaria is the main cause of paediatric consultations and hospitalisation, with an average prevalence of 18% in symptomatic patients. In patients with a *Plasmodium* infection, severe malaria represents 20% of *P. falciparum* infections [7,8].

Currently, malaria diagnostic methods are based on the identification of plasmodium or plasmodium proteins in the blood using rapid diagnostic tests (RDTs) and microscopy. RDTs are widely used in the field, with a similar or lower sensitivity compared to microscopy [9,10]. RDTs have become widely used diagnostic tools in areas with limited infrastructure. In more recent years, the use of molecular techniques that are more sensitive than microscopy has increased. These molecular techniques that amplify parasite DNA using polymerase chain reaction (PCR) are routinely applied in clinical studies and epidemiological surveys for the detection and monitoring of microscopic and submicroscopic plasmodium infections [11]. It has been demonstrated that PCR is a powerful tool for detecting submicroscopic infections in the blood, which are encountered in symptomatic as well as in asymptomatic individuals. With the aim to eliminate malaria, an ideal diagnostic method should be rapid, simple to perform, inexpensive, sensitive, accurate and non-invasive. A microscopic examination (the gold standard for malaria diagnosis and enumerating *Plasmodium* parasites) relies on demonstrating the presence of asexual erythrocytic stages from peripheral blood. Light microscopy requires blood obtained from a finger prick or via venipuncture. However, this technique (i) presents various risks and limitations in the course of repeated measurements, (ii) it is invasive, (iv) it requires strict aseptic precaution and (v) it can be tedious when patients fail to cooperate, especially in cases in which infants and children are involved [12].

An alternative method of diagnosis using samples such as stools and saliva will be of immense importance to properly investigate malaria using molecular tools. These bodily fluids can be non-invasively obtained from patients. Indeed, studies suggest that saliva and stool samples from *P. falciparum*-infected patients contain traces of parasite DNA, which can be amplified with polymerase chain reaction (PCR) [13,14]. In a study led in Uganda, a non-invasive detection method for the amplification of *Plasmodium* DNA using real-time PCR analyses of ethanol-preserved faeces showed satisfactory diagnostic performance in the estimation of malaria prevalence compared to the use of dried blood spot samples [15]. Another study carried out on Cameroonian individuals with a *P. falciparum* infection showed that stored saliva DNA in combination with ultra-sensitive qPCR assays based on the detection of the *P. falciparum* mitochondrial *cox3* gene could potentially enhance the routine testing of *P. falciparum* during disease surveillance, monitoring and evaluation of interventions for malaria elimination [16]. The use of ultra-sensitive qPCR for the detection of *P. falciparum* infection in saliva has a sensibility of more than 90%. [14] In Gabon, particularly in Franceville, no study has been carried out to assess the capacity to detect plasmodial infection from faeces and saliva. Additionally, it has been demonstrated that ultra-sensitive nested PCR assays targeting the subtelomeric variant open reading frame (*STEVOR*) gene is a powerful tool to detect *P. falciparum* infection only, with a sensitivity and specificity rate of 100% in both symptomatic and asymptomatic individuals [5,17,18]. The *STEVOR* gene is encoded in 40 copies in the subtelomeric region of the *P. falciparum* genome. This study compares the ability to detect *P. falciparum* infections in blood, saliva and stool samples using highly sensitive qPCR assays targeting the *STEVOR* gene.

## 2. Materials and Methods

### 2.1. Study Design and Population

This cross-sectional descriptive study was conducted at the paediatric ward of the “Hôpital de l’Amitié Sino-Gabonaise de Franceville” from 15th March to 31st December 2021. Franceville is a semi-urban region of southeast Gabon (1°37′15″ S, 13°34′58″ E) where *P. falciparum* malaria is highly endemic with a perennial mode of transmission and some seasonal fluctuations [19]. During this study, children aged 0–180 months with fever (tympanic temperature ≥ 37.5 °C) or with a history of fever within 24 h prior to the consultation were screened for malarial parasites, and their temperatures and haemoglobin levels were checked. To this purpose, 2–5 mL of blood was collected in EDTA Vacutainers^®^ (Becton Dickinson, Meylan, France) and analysed in a laboratory. All samples were coded with alphanumerical identifier to protect the identity of each participant. Only children with (i) uncomplicated malaria, (ii) monospecific *P. falciparum* infection and (iii) written informed consent from their parents or tutors were included in this study. This study was approved by the National Ethics Committee of Gabon (PROT 0020/2015/SG/CNE). 

### 2.2. Malaria Diagnosis

The diagnosis of malaria was performed using the Rapid Diagnosis Test (RDT) “Optimal-IT^®^-Biorad” following the manufacturer’s instructions. Optimal-IT^®^ is a direct test that detects the presence of the pLDH enzyme, and it also allows for the differentiation of specific infections by *P. falciparum* from infections with other *Plasmodium* species such as *P. vivax*, *P. ovale* and *P. malariae*. All results were confirmed via microscopy by two experienced microscopists. The parasite load was, therefore, determined and expressed as the number of parasites (asexual forms) per microliter of blood using Lambaréné’s method [20]. Consequently, children with uncomplicated and monospecific *P. falciparum* malaria infection were asked to provide saliva and stool samples. 

### 2.3. Saliva and Stool Collection

Saliva was collected using sterile swabs (~200 µL), and a few milligrams of stool was collected in sterile pots containing a conservation medium (RNALater^®^, QIAGEN) [21] and stored at −70 °C until DNA extraction. Each child observed with malaria infection was offered a 3-day course of artemether/lumefantrine (Coartem^®^ Lonart; Cipla, Mumbai, India). 

#### DNA Extraction

DNA template was extracted from (i) blood using the DNA Blood Omega Bio-Tek E.Z.NA^®^ kit (Omega Bio-Tek), Inc.400 Pinnacle Way, Suite 450, Norcross, GA 30071, USA) (ii) saliva using the Qiagen^®^ saliva DNA extraction kit (Qiagen, 19300 Germantown Rd, Germantown, MD 20874, USA) and (iii) stool using the Qiagen^®^ stool DNA extraction kit (Qiagen, 19300 Germantown Rd, Germantown, MD 20874, USA) following the manufacturer’s instructions. Due to the difficulty in obtaining DNA in stool samples and in order to check the quality of the extracted DNA, we quantified the DNA template with nanodrops only in the stool samples.

### 2.4. GAPDH qPCR Amplification

In order to check the presence of DNA and quantify it in all matched samples, we amplified the human GAPDH housekeeping gene using quantitative real-time PCR (qPCR). qPCR was performed using the following primers: GAPDH_F: 5′-GAAGGTGAAGGTCGGAGT-3′; GAPDH_R: 5′-GAAGATGGTGATGGGATTTC-3′. The probe was 5′-rox-CAAGCTTCCCGTTCTCAGCC-BHQ2-3′, and the conditions were 94 °C for 10 min, 94 °C for 30 s, 55 °C for 30 s and 72 °C for 30 s for 40 cycles in the presence of the Universal PCR Master mix kit (Apply Biosystems). 

### 2.5. Nested qPCR Targeting STEVOR Gene

Nested real-time PCR assay targeting a conserved fragment of STEVOR gene was used. Only blood, saliva and stools from infected children (with written informed consent) were tested.

Nested qPCR of saliva, nested qPCR of stools and nested qPCR of blood were performed in the same conditions. A total of 2.5 µL of DNA template (extracted from blood, saliva and stools) were amplified using a Perkin Elmer thermal cycler. The total volume of reaction was 25 µL containing 1× PCR buffer as supplied by the manufacturer; 17 µM each of dATP, dCTP, dGTP and dTTP; 0.75 pM of each primer, P5, P18, P19 and P20 (Table 1) and a 0.625 unit of Taq DNA polymerase. For the second round of amplification, 1 µL of the first PCR product was used as template in a qPCR reaction (20 µL final volume) containing 0.75 µM of each primer, P17 and P24 (Table 1), and 1× Power SYBR Green Master Mix (Applied Biosystems). 

All the amplifications were performed in 96-well optical PCR plates using an Applied Biosystems 7900 HT Fast Real-Time PCR system and under the following conditions: 93 °C for 3 min and 35 cycles of 93 °C for 50 s, 50 °C for 50 s and 72 °C for 50 s.

### 2.6. Data Analysis

Data were analysed using R software version 3.5.3 (manufacturer) [22]. Qualitative variables were described by proportion and quantitative variables were described by mean and standard deviation (SD). The proportions of qualitative variables were compared using the non-parametric Chi-squared test or Fisher’s exact test for numbers below 5. Student’s *t*-test was used for comparison of two means, and non-parametric Kruskal–Wallis test was used when the numbers were insufficient. The level of significance was assumed at *p* < 0.05.

## 3. Results

During this study, there were 367 children aged 1 to 180 months who were screened for malaria infections. The febrile patients included in the analysis are shown in Figure 1.

The numbers of males and females were 201 (54.7%) and 166 (45.23%), respectively. The mean temperature was 38.18 ± 1.17 °C. Table 2 shows the demographic and clinical characteristics of all the patients.

The prevalence of malaria calculated via microscopy was 21.80% (80/367; 95% CI, 17.88 to 26.30), with a mean parasitaemia of 26,512.23 ± 4687.12 parasites/µL (ranging from 121 to 201,600 parasites/µL). *P. falciparum* was responsible for 100% of the infections. 

A total of 35 patients agreed to participate in the study, and each patient provided a stool and saliva sample. The mean values of their haemoglobin and temperature were 9.03 ± 1.75 g/dl (95% IQR [8.25–10.27], min = 3.20 g/dl and max = 11.9 g/dl) and 37.1 ± 6.54 °C (95% IQR [37.3–39.2], min = 35.9 °C and max = 40 °C), respectively. To validate the quality of the extracted DNA, the GAPDH gene was amplified. The results show that DNA was detectable in all of the samples. 

To determine the performance of the nested qPCR of saliva, the nested qPCR of stools and the nested qPCR of blood, thick blood smears were used as the gold standard. We found that the sensitivity of the qPCR of blood was the highest compared with the qPCR of saliva and the qPCR of stools (*p* < 0.001). Table 3 shows the different sensitivities for the qPCR assays.

To determine the impact of parasitaemia on the performance of the nested qPCR of saliva and the nested qPCR of stools, the data were stratified into different parasitaemia, and the sensitivity of the PCR assay was calculated for each group (Table 4).

There were 27 false negative results when saliva was used to perform a qPCR, and relatively more false negative results (30) when stools were used. False negative results were observed in all ranges of parasitaemia (Table 4), and the sensitivity levels of both the qPCR for saliva and the nested qPCR for stools were extremely variable with regard to these ranges of parasitaemia.

We found that the Cycle Threshold (Ct) value of the blood samples’ amplifications was significantly lower than the stool and saliva samples’ Ct values (respectively, 2.7 × 10^−5^ and 5.2 × 10^−6^). There was no statistical difference between the CT values of the faeces and saliva samples (*p* > 0.06). 

## 4. Discussion

This study found that the percentage of children who went to a hospital with a fever and who tested positive for malaria using microscopy was around 22%. Similarly, previous studies led in 2011 and 2016 reported the same proportion [4,23], indicating that the malaria infection rate remains stable in Gabon (Franceville). 

Using thick blood film microscopy as the gold standard, we compared the sensitivity of nested qPCR assays targeting the STEVOR gene in detecting *P. falciparum* in blood, saliva and stool samples. We showed that *P. falciparum* was detected with sensitivity rates of 100%, 22.86% and 14.29%, respectively, in blood, saliva and stool samples. This allowed for speculations on the concentration of *P. falciparum* DNA in saliva and stool samples. Indeed, the amplification of the GAPDH gene in each sample showed that the DNA extraction was successful, and the analysis showed that there is more DNA in blood samples than in saliva and stool samples, which is coherent with the findings of previous studies [13,24,25]. In the same line, the *P. falciparum* DNA concentration may be considerably reduced or degraded in saliva or stools, rendering its amplification difficult even with a highly sensitive technique such as PCR. Further investigation is required.

Reports have indicated that saliva could be a potential non-invasive alternative to blood for malaria diagnosis [13,14,26,27]. The results presented herein show that *P*. *falciparum* was detected in saliva with a sensitivity of 22.8%. This is relatively low compared to previous findings [14,26]. The mean reason for this difference could be explained by the targeted gene choice. Indeed, in the cited articles, different genes were targeted (subunit ribosomal RNA gene and multicopy 18 s rRNA plasmodial gene). This result could also be explained by the saliva collection and storage processes. Indeed, it has been shown that when saliva is collected in absolute ethanol [24], the sensitivity of parasite detection increases. This was not the case in our study. In addition, we extracted DNA from less than 1 mL of saliva, whereas other researchers have used 2 mL to boost their chances. Additionally, the STEVOR gene amplification approach was reported to be more sensitive [17] than the 18 s rRNA plasmodial gene amplification approach. Furthermore, it was demonstrated that, with a parasitaemia of >10,000 parasites/µL in blood, the sensitivity of the nested PCR of saliva targeting the 18 s rRNA plasmodial gene is 100% [14]. Interestingly, in our study, the sensitivity of the nested qPCR targeting the STEVOR gene was not influenced by parasitaemia. A more in-depth study is needed to better explain the mechanisms hidden behind such observations.

The majority of studies investigating the detection of plasmodium DNA in stools are focused on monkeys [28,29,30]. Only few studies have explored the presence of *P. falciparum* DNA in human faeces. To date, the mechanisms whereby *Plasmodium* DNA is shed into human stools are not well understood. It is believed that excretion into the biliary system, as it has been proposed for *P. yoelli*-infected mice, could be a plausible explanation [28]. Our results show that *Plasmodium* DNA was detectable in human stools (sensitivity of 14.29% using PCR). This extremely low sensitivity, compared to the one obtained from the blood samples (100%), could be explained by the large amount of degraded parasite DNA in the stool. Indeed, the literature states that infected erythrocytes are degraded in both the spleen and liver during the course of infection and a large amount of degraded parasite constituents including DNA are produced in the liver. The degraded constituents are excreted from the liver (via the bile) into the faeces [28]. Furthermore, the stool storage method could be an additional reason for such a low *Plasmodium* DNA detection. Indeed, in another study led in Uganda, they kept stools in alcohol prior to performing DNA extraction [15]. In this study, the detection of parasite DNA in stools was higher than in another study led in Cameroon (3%) [31]. This difference may be due to the fact that they worked with asymptomatic subjects with no signs of clinical illness, whereas we included only malaria-positive subjects.

For some faecal samples, a qPCR assay was at the limit of detection (≥40 Ct), and the Ct value of blood was statistically lower than the saliva and faeces values, which means that there is more parasite DNA in blood than in saliva and stools. This last result is obvious because the parasite infects red blood cells, and it is more abundant in blood than in any other fluid.

## 5. Conclusions

*Plasmodium falciparum* DNA is detectable in saliva and stools using nested PCR targeting the STEVOR gene, although the sensitivity is considerably reduced when compared to the results obtained from blood samples. However, the small sample size of this study means that it is not possible to draw any firm conclusions about the suitability of these non-invasive samples in detecting malaria parasite DNA in saliva and stools. Further studies are needed to improve the detection of *Plasmodium* DNA in these fluids. 

## Figures and Tables

**Figure 1 diagnostics-13-03271-f001:**
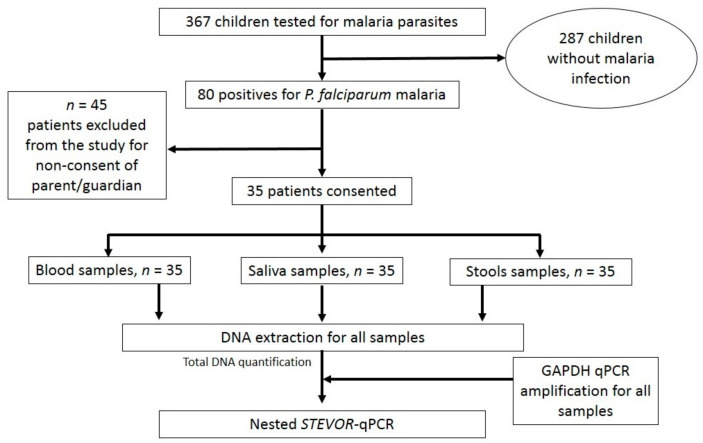
Flowchart of enrolment for patients, sample collection and laboratory assays.

**Table 1 diagnostics-13-03271-t001:** Nucleotide primers used in STEVOR gene amplification.

Primers	
P5	5′-GGG AAT TCT TTA TTT GAT GAA GAT G-3′
P17	5′-ACA TTA TCA TAA TGA (C/T)CC AGA ACT-3′
P18	5′-TTT CA(C/T) CAC CAA ACA TTT CTT-3′
P19	5′-AAT CCA CAT TAT CAC AAT GA-3′
P20	5′-CCG ATT TTA ACA TAA TAT GA-3′
P24	5′-GTT TGC AAT AAT TCT TTT TCT AGC-3′

**Table 2 diagnostics-13-03271-t002:** Biological characteristics of patients.

	Value Parameter
***N* (number)**	367
**Hemoglobin (g/dL)**	
- Mean (SD)	8.81 (2.12)
- Median [IQR]	9.03 [8.20–10.3]
**White blood cells (×10^3^/µL)**	
- Mean (SD)	5.80 (3.65)
- Median [IQR]	4.90 [4.05–5.65]
**Red blood cells (×10^3^/µL)**	
- Mean (SD)	3.86 (0.69)
- Median [IQR]	3.78 [3.54–4.44]
**Platelets (×10^3^/µL)**	
- Mean (SD)	185 (107)
- Median [IQR]	162 [100–254]

**Table 3 diagnostics-13-03271-t003:** Sensitivity of nested qPCR for saliva, stool and blood samples.

	Number	Sensibility [95%CI]
qPCR-based assay for blood		
Negative	0	100%
Positive	35	
qPCR-based assay for saliva		
Negative	27	22.86% [12.07–39.02]
Positive	8	
qPCR-based assay for stools		
Negative	21	14.29% [6.26–41.59]
Positive	5	

**Table 4 diagnostics-13-03271-t004:** Sensitivity of both nested qPCR for saliva and nested qPCR for stools with regard to parasitaemia levels.

Parasitaemia/µL	*n*	Nested qPCR Assay Characteristics					
		True Positive		False Negative		Sensitivity (in %)	
		Saliva, *n* (%)	Stool, *n* (%)	Saliva, *n* (%)	Stool, *n* (%)	Saliva	Stool
<1000	5	3	0	2	5	60	0
1000–10,000	15	2	3	13	12	13.33	20
10,001–50,000	10	2	1	8	9	20	10
50,001–100,000	3	1	0	2	3	33.33	0
>100,000	2	0	1	2	1	0	50

## Data Availability

Data can be shared as needed.

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
