# Peer review of "Detection of Plasmodium falciparum in Saliva and Stool Samples from Children Living in Franceville, a Highly Endemic Region of Gabon"

_diagnostics, 2023, doi:10.3390/diagnostics13203271_

Round 1

Reviewer 1 Report

Diagnostics-2433767-peer-review-v1-comments

Detection of Plasmodium falciparum in both Saliva and stool  samples from children living in Franceville, a highly endemic  region of Gabon

Roméo Karl Imboumy-Limoukou et.al.

Overall Comments :

The paper presents a study aimed at assessing the suitability of saliva, stool and blood samples for the detection of P. falciparum infection in children in Gabon using a qPCR STEVOR-gene assay. The study builds on the previously published reports showing saliva and stool from malaria patients contain Plasmodium DNA. The study has been conducted by enrolling three hundred sixty-seven (367) children aged 1 to 180 months in the first instance, of which 80 were confirmed positive for Pf infection, and of these, only 35 subjects consented to provide blood, saliva, and stool samples for the study. The study used microscopy as the gold standard and only thick smears were used for the diagnosis of malaria and parasite quantification. The results showed that compared to microscopy, qPCR targeting the STEVOR gene had a sensitivity of 100%, 22.86%, and 14.29% 28 using blood, saliva, and stool samples, respectively for the detection of Plasmodium falciparum. The study concludes that qPCR STEVOR-gene for detecting Plasmodium DNA in saliva and stool cannot be considered an alternative to current malaria detection techniques using blood specimens.

The introduction provides sufficient background for the study but there are issues related to the language and expression which need to be addressed.

Material and methods require revision to make them comprehensive and to remove ambiguity. The authors report it to be a transversal/cross-sectional study but the study period is missing. Minimal details are available regarding the study population of 35 children finally included in the study.  Malaria diagnosis and parasite quantification have been done using an old method of Lambaréné(2001) while newer and latest WHO techniques are recommended and expected to be followed as per the WHO guidelines. It is not clear why the authors have chosen this old method. References of the methods followed for DNA amplification/PCR protocols, if any, should be cited. In case the authors used a novel approach, it should be clearly indicated. Though other details provided in the material and methods would suffice for this paper, the language and expression issues need to be addressed in this part of the paper also for better clarity and to ensure that the intended meaning is conveyed.

In discussion as well as elsewhere in the paper, the PCR method used in other studies has been referred to but not specified.  Since studies cited for comparison in the discussion probably did not use the same gene and PCR method,  it may not be feasible to compare two different methods on a one-on-one basis without specifying the target genes used in other studies. This specific part needs modification and revision for better clarity in the introduction as well as the discussion.

In conclusion, the authors generalize the outcome of this study to Plasmodium spp. Observations and results of this study should conclude only Plasmodium falciparum and not generalize to all Plasmodium spp. as this study did not cover other Plasmodium spp. Accordingly, specific areas need modification in the abstract also and where ever applicable in other parts of the paper. Lastly, scientific names like P. falciparum need to be italicized in the entire paper.

The word both may be removed from the title.

Specific comments for each section of the paper are given below for guidance on specific areas requiring revision.

Specific Comments :

1.     Title: It is recommended to remove ‘both’ from the title.

2.     Abstract :

Revision is needed to address the issues related to English grammar and language. Revision is required to address the issues related to methodology and discussion as detailed in these sections. The overall revision of the abstract after incorporating suggested changes in the paper is recommended to ensure uniformity.

3.     Introduction:

· In addition to the above suggested revisions please review the following sentences for clarity :

·   Line nos. 37-38,47-48,65-67,82-83

· Please consider replacing the word ‘reveal’ with ‘detect or diagnose, as appropriate, in the entire text.

4.     Materials and Methods :

· Lines 104-118: These are a part of instructions to the authors by the journal and are not required to be reproduced in the paper. Please delete. A brief introduction instead should suffice.

· Please incorporate the study period and clearly mention the type of study done.

· Line 141: Reference of  Lambarene's method dates back to 2001. It is advised to follow the WHO quality assurance manual 2016 for all malaria microscopy-related work in future. The revised WHO SOPs for malaria microscopy should also be referred to in future. For this paper, the authors may compare the method used by them with the suggested WHO methods and highlight the differences, if any, along with a suitable explanation for choosing this old method over the revised and latest ones recommended by WHO.

· Lines 147-148,156: Check for clarity and intended meaning.

· Line 170: two point five should be preferably written as 2.5 to maintain a uniformity of expression in the paper.

·        Lines 173 & 176: The relevance of referring to Table 1 is not clear. Please check.

·Line 181: Please write the name of the programme used for data analysis followed by the reference(22)

· What does (the manufacturer) indicate for R software version 3.5.3? I am not sure if this is required. Please check this.

Results:

·Figure 1: Check language, grammar and expression to ensure that the intended meaning is conveyed e.g., screen, No malaria infection, No consented etc.

·Table 1: The heading reads as ‘Biological characteristics of patients’, then there is another sub-heading ‘General characteristics’. Pl check what exactly needs to be conveyed.

·Line 197: Please correct ‘screen children’

·Line 201-202: Please add the CI or median IQR  also to know the lowest and highest values of haemoglobin and temperature for the children finally included in the study.

·        Table 2: Revise. Suggestion: add samples to the title for clarity.

Discussion :

·        Lines 222-228:  This is a part of the instructions to authors and not needed to be reproduced in the paper. Please delete.

·        Line 238-243: Was the DNA quantified? If, yes, please add a table in the results or a photograph to qualify the statement. Otherwise, an appropriate explanation justifying the statement may be added.

·        Lines 244-247: Not clear. Did the studies cited for comparison in the discussion use the same gene and PCR methods? If not, is it feasible to compare two different methods on a one-on-one basis? Please modify for better clarity, where ever feasible.

·        Line 247-248: Please elaborate on the line 'This could be explained by the saliva collection and storage processes’. The intended meaning is not clear.

                      Conclusion:

Please restrict the conclusion to only Plasmodium falciparum and do not generalize to all Plasmodium spp. as this study did not cover other Plasmodium spp.

Please ensure that necessary changes to this effect are made in the entire paper including the abstract.

References :

Reference no. 1 needs to be cited appropriately.

Ensure that the journal format is followed for all references.

Please add DOI to all references, where ever available, to keep up with the latest publication trends.

English language needs thorough review and revision for expression and grammar to ensure that the intended meaning is conveyed.

Many language issues exist as of now in the paper.

Author Response

Cf PDF

Reviewer 2 Report

Imboumy-Limoukou and colleagues tested the suitability of three different sample types (Blood, saliva and stool) for the molecular detection of Plasmodium falcifparum infection using nested qPCR targeting STEVOR-gene in a cohort of malaria patient in children of Gabon. While P. falciparum can be readily detectable (100%) in thick blood smear and blood DNA, detection rates in saliva (22.86%) and stools (14.29%) were compromised. Malaria is blood protozoa and thick blood smear is the gold standard in malaria diagnosis. Although, non-invasive sampling and molecular detection has been investigated recently with variable findings.

Major issues:

1.      The study lacks significant novelty as similar research have been conducted elsewhere (reviewed by the author in line 81-102).

2.      The sample size (n = 35) seems too small to draw a conclusion about the suitability of these non-invasive samples in detecting malaria parasite DNA in saliva and stools.

3.      Results: PCR inhibitors present in the feces could affect the amplification of the malarial DNA. Although, author used GAPDH as DNA QC, no such Ct data was provided. Besides, author mentioned the volume (2.5 ul) of DNA used in the PCR, but it was not mentioned if they used similar quantity (in ng) for all three sample types. In line 205, author stated that “We found that the sensitivity of PCR was the highest (p<0.02)”, however, so such statistics or comparison was shown in the results.

Other issues:

1.      Line 108-118 should be removed.

2.      Table 1 showing the list of primers is missing.

3.      Data analysis: Do statistical tests (chi-square or student T test) data are presented.

4.      Line 196 (Table 1): What the author meant by “Mean age”? Please correct x103/μl

5.      Discussion: Please remove line 222-225.

Needs some minor corrections only.

Author Response

Cf PDF

Round 2

Reviewer 2 Report

All comments were addressed. 

Minor editorial corrections are necessary.